# The Novel Concept of Synergically Combining: High Hydrostatic Pressure and Lytic Bacteriophages to Eliminate Vegetative and Spore-Forming Bacteria in Food Products

**DOI:** 10.3390/foods13162519

**Published:** 2024-08-12

**Authors:** Dziyana Shymialevich, Michał Wójcicki, Barbara Sokołowska

**Affiliations:** Department of Microbiology, Prof. Wacław Dąbrowski Institute of Agricultural and Food Biotechnology—State Research Institute, Rakowiecka 36 Str., 02-532 Warsaw, Poland; michal.wojcicki@ibprs.pl (M.W.); barbara.sokolowska@ibprs.pl (B.S.)

**Keywords:** HHP, virulent bacteriophage, hurdle technology, biocontrol, biopreservation, spore-forming bacteria

## Abstract

The article focuses on the ongoing challenge of eliminating vegetative and spore-forming bacteria from food products that exhibit resistance to the traditional preservation methods. In response to this need, the authors highlight an innovative approach based on the synergistic utilization of high-hydrostatic-pressure (HHP) and lytic bacteriophages. The article reviews the current research on the use of HHP and lytic bacteriophages to combat bacteria in food products. The scope includes a comprehensive review of the existing literature on bacterial cell damage following HHP application, aiming to elucidate the synergistic effects of these technologies. Through this in-depth analysis, the article aims to contribute to a deeper understanding of how these innovative techniques can improve food safety and quality. There is no available research on the use of HHP and bacteriophages in the elimination of spore-forming bacteria; however, an important role of the synergistic effect of HHP and lytic bacteriophages with the appropriate adjustment of the parameters has been demonstrated in the more effective elimination of non-spore-forming bacteria from food products. This suggests that, when using this approach in the case of spore-forming bacteria, there is a high chance of the effective inactivation of this biological threat.

## 1. Introduction

Most recorded food poisonings are caused by the consumption of microbiologically contaminated raw materials and products, including fruit, vegetables, or seafood. Currently, the dietary recommendations provide an increasing emphasis on healthy eating and the consumption of wholesome and minimally processed food. The use of high-temperature pasteurization does not preserve the fresh appearance and taste of the product, which is less accepted by consumers [1]. For this reason, modern physical, chemical, or biological methods are being sought to completely or partially replace high-temperature pasteurization. In recent years, the food industry has devoted much attention to the development of non-thermal high-hydrostatic-pressure (HHP) processing technologies. When using the HHP technique, no chemical preservatives are required. Additionally, preservation can be performed at ambient or low temperatures, eliminating the energy consumption for heating and cooling. For this reason, it is an environmentally friendly processing technology [2]. An important aspect is that food products are in the final packaging and do not have direct contact with the equipment, which prevents secondary contamination.

Another developing method of food preservation is biocontrol using a cocktail of lytic bacteriophages. The elimination of bacterial pathogens is highly effective due to the specificity of bacteriophages, which is limited to the ability to infect only the target species. The ability to self-replicate makes a small dose of the cocktail effective in biopreservation. Despite the legislative restrictions in many countries (including European Union Member States), phage biocontrol is increasingly accepted by society as a natural and green biotechnology. By implementing the Farm To Fork Strategy, the heart of the European Green Deal aiming to make food systems fair, healthy, and environmentally friendly, bacteriophages can guarantee the production of health-promoting and sustainable food. Due to the possibility of bacteriophages infecting only live and active bacterial cells, their effectiveness against spores is limited due to the lack of specific receptors, their morphology, and the metabolic activity of the spores. However, bacterial spores are ubiquitous in water and soil, which is why they often enter food, affecting consumer safety. The difficulty in eliminating them lies in their high tolerance over wide ranges of temperatures, pH values, water activity, or industrially applied pressure values (600 MPa) [3]. In recent years, several studies have been conducted and proved that the use of higher-pressure treatment parameters with simultaneous heating (above 70 °C) of the product leads to the significant inactivation of the spores [4]. However, this approach causes changes in the sensory and nutritional characteristics of the product, which is not acceptable to consumers. A solution may be to use pressure to activate the spores. The resulting vegetative forms will then be eliminated using a cocktail of strictly lytic bacteriophages applied to the food product without altering the sensory properties of the product.

This article aims to elucidate the novel concept of synergically combining HHP and lytic bacteriophages to eliminate vegetative and spore-forming bacteria from food products effectively. By presenting the current state of knowledge and the potential benefits of this innovative approach, this article seeks food safety standards and minimizing dependence on the traditional thermal processing methods.

## 2. High Hydrostatic Pressure—Innovative but Not New Technology

The concept of using high-pressure technology in food preservation is not new. The first mention of its use dates back to 1899 from the United States of America (USA), when Bert Hite observed that after applying a pressure of 650 MPa for 10 min at room temperature, the durability of milk increased. However, it took almost a century, when in 1980 the first products such as yogurts, jams, and jellies were treated with HHP and introduced to the Japanese market. Currently, products preserved by high pressure are increasingly found on store shelves [5].

HHP treatment is an innovative food preservation process used commercially for non-thermal pasteurization. The HHP technique involves the use of ultra-high pressure (200–800 MPa) on food products that are hermetically sealed and thermally isolated [6]. The treatment is carried out in continuous or semi-continuous mode—for liquid products, both modes are used, while for solid products, only batch mode is used. The finished product is placed in a special pressure chamber, which is filled with a pressure-transferring agent. The pressure is transmitted through a liquid medium (usually water but also polypropylene glycol, silicone oil, and castor oil), which ensures uniform pasteurization and immediate achievement of high pressure parameters [1]. High pressure at room temperature during the food preservation process causes a moderate increase in temperature (adiabatic heating). Depending on the composition of the food products, the temperature increases by 3–6 °C for every 100 MPa. After decompression, the temperature of food products decreases to the initial temperature. During the pressurization process, heat is neither stored nor transferred [7]. The uniform distribution of pressure in a closed sample is governed by two basic principles: the isostatic principle and Pascal’s principle. According to the isostatic principle, pressure that is applied to a liquid medium in a closed environment exerts equal pressure on objects at any point in the environment, regardless of the shape or size of the object. According to Pascal’s principle, a change in pressure causes an external force to be uniformly applied to the fluid at rest in a closed container and is transferred without losses to the walls of the container [8]. For this reason, the shape and size of the packaging do not affect the pasteurization process, and what is more, products of different dimensions can be processed in one batch [2].

In recent years, research interest in the influence of hydrostatic pressure on the quality, safety, and microbiological stability of foods has increased significantly [9]. It has been shown that HHP affects only non-covalent bonds (hydrogen, hydrophobic, ionic bonds) [10], and low-pressure values do not cause significant changes in the component responsible for color, smell, and bioactive compounds, which means the products retain most of their freshness features [11]. However, after HHP, color changes occur in some products (e.g., meat) [1]. Some studies suggest that the microbiocidal effectiveness can be increased by using gentle heating or the addition of biopreservatives such as nisin [12].

### 2.1. Mechanisms of Bacterial Cell Damage under the Influence of HHP

#### 2.1.1. Injury to Vegetative Bacterial Cells

Currently, there are numerous studies describing the inactivation of vegetative cells that cause food spoilage using the HHP method. Mostly, the kinetics of inactivation are characterized by a constantly decreasing number of bacteria in the product or model system. However, significant differences in pressure resistance have been demonstrated within strains of the same species [13]. In terms of sensitivity to high pressure, microorganisms are divided into three groups: Gram-negative bacteria, which are inactivated at 300 MPa and above, fungi (yeasts and molds) at 400 MPa, and Gram-positive bacteria inactivated at 600 MPa and above [14]. Bacterial spores are exceptionally resistant, which, when using commercial doses of HHP, can be destroyed only in the germination phase. Bacterial spores, like viruses, can be inactivated using very high pressure (more than 1000 MPa) and simultaneous heating. In the case of vegetative cells, the denaturation and inactivation of proteins occur after applying HHP above 300 MPa. Meanwhile, after applying a pressure of 200 MPa and above, the disruption of the oligomeric structures was observed, which consequently led to the dissociation of monomers. The occurrence of barotolerant cells that have developed a resistance mechanism due to the adaptation to stress and selection has also been reported [15]. The differences in pressure resistance also depend on the shape of the bacteria. Research conducted by Pilavtepe-Çelik et al. [16] showed that Gram-positive facultative anaerobe cocci (*Staphylococcus aureus*) are the most baro-resistant, while Gram-negative rod-shaped bacteria (*Escherichia coli*) are more resistant compared to slender Gram-negative rod-shaped bacteria (*Pseudomonas aeruginosa*). The bacterial cell damage caused by HHP is associated with various mechanisms that can be divided into 14 types (Figure 1).

At sufficiently high pressures, changes in the cell structure and the disruption of numerous metabolic processes are observed. The lipid membrane is particularly sensitive to the effects of HHP due to its high compressible potential. Under the influence of pressure, the lipid layer loses fluidity and becomes insoluble, and protein–lipid interactions weaken [17]. Differences in resistance between bacteria are caused by the chemical composition and structural properties of the cytoplasmic membranes of both Gram-positive and Gram-negative bacteria. The phospholipid bilayer is closely packed during the compression phase, facilitating the transition to the gel state, and, during decompression, the bilayer structure is disrupted, leading to the formation of pores and the leakage of the cytoplasmic content. For the bacterium to function, the membrane must maintain a fluid state [18]. This condition is determined by the composition and percentage of unsaturated fatty acids (FAs). It was observed that barophilic and barotolerant bacteria are characterized by a higher FA content in the membrane. The transition from saturated to unsaturated FAs as a result of adaptation to the applied pressure has also been described [19].

Biochemical processes are a key element in maintaining the life processes of a bacterial cell. High-pressure treatment leads to the denaturation of the functional proteins, which limits the movement of protons and leads to a decrease in intracellular pH. Additionally, HHP causes irreversible protein aggregation when high parameters are used. Studies show that *E. coli* mutants lacking superoxide dismutase (SOD) as a reactive oxygen species (ROS) scavenger are more sensitive to high pressure compared to the wild type [20]. This phenomenon indicates an inactivation mechanism involving oxidative stress. However, the strain lacking catalase was more stable after HHP, which indicates no effect of H_2_O_2_ [21]. Another important effect of bacterial cell death under the influence of high pressure is the inhibition of protein translation due to the breakdown of ribosomes into their subunits (70S → 30S + 50S). In the case of *E. coli*, the application of a pressure of 60–70 MPa has been shown to disrupt the association and dissociation of the 70S ribosome [22]. By increasing the pressure, the genetic material also changes [23]. As described above, high pressure disrupts the action of the enzymes on which the functionality of the genetic material depends. Due to the inactivation of the enzymes necessary for DNA replication and transcription, these processes are limited. As a consequence, the genetic material condenses. Covalent and hydrogen bonds are more resistant to pressure compared to hydrophobic, ionic, and electrostatic interactions. Due to the hydrogen bonds in the DNA helix, the nucleic acid appears to be more stable. However, under the influence of high pressure, the endonuclease cleaves the DNA, which leads to its condensation [24]. Moreover, research indicates that a pressure of 50–100 MPa leads to the inhibition of microbial DNA replication and transcription.

The above-mentioned mechanisms of bacterial cell inactivation after HHP application were observed during the imaging of *E. coli* and *Listeria innocua* strains using epifluorescence microscopy (EFM), scanning electron microscopy (SEM), and transmission electron microscopy (TEM). The loss of the overall cell shape (i.e., collapse, flattening, and compressed envelopes), aggregation and empty spaces of the cytoplasm, and significant disorganization of nucleic acid were demonstrated. The deprivation of the integrity of the cytoplasmatic membrane also resulted in the efflux of the cytoplasmic content into the extracellular space [25].

The use of HHP to inactivate bacteria may also change the appearance of their colonies on an agar plate or cause them to appear later on the microbiological agar due to the time needed to regenerate sublethal damage. Regenerated colonies are often more heterogeneous compared to untreated HHP controls, suggesting miscellaneous damage to the bacterial population [26]. Based on observations, cells can be damaged to varying degrees. Bacterial damage can be divided into sublethal and lethal [25]. A sublethally damaged cell is active, and, after the regeneration and adaptation to new environmental conditions, it may show the characteristics of a healthy cell. An example would be a product stored in inappropriate conditions, where bacterial growth can be observed, which results in product spoilage and a potential threat to the consumer’s health. The rapid proliferation of this type of injury results in bypassing the repair of the damage, resulting in bacterial cell death. Stopping division without a stress factor can lead to the repair of metabolism. If the harmful factor acts again on such a damaged cell, it is completely damaged, which is advantageous in the case of combined methods. Lethal damage is irreversible, cannot be repaired, and causes immediate cell death [27]. Table 1 presents selected studies on the reduction in bacterial load after HHP treatment, at various pressure, time, and temperature parameters, both in model media and food matrices. 

An effective approach to eliminating microorganisms is the use of mild heat stress after HHP treatment. The results suggest that such damage to bacterial cells causes lethal inactivation. Lower pressure requires longer treatments or temperature combinations, both of which increase the production costs. It is worth emphasizing that the elimination of some microorganisms can also be achieved at negative temperatures [21]. In the case of endogenous factors, the pH value, water activity, and the concentration of ionic and non-ionic solutes have a significant impact on cell inactivation. The best results of high-pressure processing are obtained for products with low pH values and high water activity. A deterioration in the efficiency of this process was observed in the case of dried products such as milk powder, spices, and dried meat products. Trapping bacterial cells in the fat or the oil phase protects their cells against the applied pressure because they constitute fractions of ingredients with low water activity value [23].

The effectiveness of bacterial inactivation largely depends on the endo- and exogenous factors occurring in parallel with the applied pressure. Stress caused by environmental conditions such as heat shock, cold shock, or osmotic stress may also cause increased tolerance to pressure. For example, *Listeria monocytogenes* under stress conditions enters a dormant phase and becomes more barotolerant [33]. It is worth emphasizing that bacterial cells that are resistant to heating are also more resistant to pressure. Moreover, the susceptibility of microorganisms to HHP depends on the physiological state of the cell. Microorganisms in the logarithmic growth phase are always more sensitive to high pressure than in the stationary phase. Furthermore, research confirms that the prophages and bacterial defense systems related to preventing the transfer of genetic material may be involved in the survival of bacteria in conditions of increased pressure [33].

Moderately damaged cells can regenerate depending on the nutrient content, pH, or temperature. Koseki and Yamamoto [34] conducted a study in which *E. coli* cells were suspended in phosphate-buffered saline (PBS) and exposed to a pressure of 500–600 MPa at various temperatures for 10 min. When stored at a refrigerated temperature (4 °C) for 120 h, no colonies were detected; however, it was observed that increasing the storage temperature to 25 °C caused the bacteria to grow. On the other hand, at 37 °C the number of microorganisms dropped again to an undetectable level [34]. An example of cell regeneration in a nutrient-rich environment is research using the bacterial pathogen *L. monocytogenes* suspended in trypticase soy broth (TSB). The strain was treated with a pressure of 500 MPa at 25 °C for 10 min. Immediately after the treatment, damaged cells were detected on the non-selective agar, but none on the selective agar. After one day of storage of the samples at 0 °C and 15 °C on selective agar, 0.5 log CFU mL^−1^ and 5.0 log CFU mL^−1^ were observed, respectively [35]. The research conducted by Nasiłowska et al. [36] indicates that carrot juice (pH 6.0–6.7) inoculated with *L. innocua* and treated with HHP of 300–500 MPa supported the growth and regeneration of the strain compared to beetroot juice (pH 4.0–4.2). Due to the presented limitations in the inactivation of microorganisms after the use of HHP, new possibilities for combining the technique with natural antimicrobials are currently being developed. It has been found that the addition of bacteriocins (e.g., nisin) or essential oils increases the microbiological safety of food [37].

#### 2.1.2. Bacterial Spore Damage

The mechanism of the inactivation of spores and vegetative bacterial cells by HHP is different, which is related to the structure of both forms of microorganisms. The resistance of bacterial spores is related to the presence of multiple protective layers, low water compactness, and the presence of small soluble proteins that protect DNA [38,39,40]. It is believed that spores are activated by moderate pressure in the range of 50 to 600 MPa, which causes them to germinate into vegetative forms [41,42,43]. This is accompanied by several changes, such as the release of calcium dipicolinate (CaDPA) and hydrolysis of the peptidoglycan (PG) by specific cortex enzymes (e.g., cortex-lytic enzyme, CLE). The resulting changes activate the water uptake and expansion of the spore core, which results in the activation of core enzymes and resumption of metabolism. The completion of these processes produces a fully germinated spore that transforms into a vegetative form of bacteria capable of further cell division [39]. The use of high temperatures in the spore germination phase can effectively eliminate them. This is associated with the release of CaDPA, which causes the spores to become temperature-sensitive and damage essential core proteins [44,45,46]. Research indicates that moderate–high-pressure (MHP) treatments (50–300 MPa) activate gerA-type germination receptors (GR), after which the spore begins to absorb water and its internal structure is reorganized, leading to its germination. Meanwhile, very high-pressure (VHP) treatments (400–600 MPa) open the SpoVA channel, leading to the release of CaDPA [47]. Most bacterial spores contain the SpoVA protein, suggesting that VHP is generally more effective. However, spores of *Bacillus* species and some *Clostridium* species contain gerA-type GR and can germinate under MHP conditions [48]. Many research centers around the world use the HHP technique to inactivate saprophytic, pathogenic, and spore-forming bacteria in food matrices. Table 2 presents selected studies on the reduction in the load of bacterial spores following HHP treatment at various pressure, time, and temperature parameters, both in the model media and food matrices.

After analyzing the above-mentioned studies, it was confirmed that bacterial spores are the most resistant to HHP. In many cases, a pressure of 600 MPa does not completely inactivate the spores. Taking into account vegetative cells, the reduction strictly depends on the matrix used and the parameters of the HHP process. However, often, a pressure of 400 MPa is sufficient for the complete reduction in the vegetative cell contamination, while there are exceptions. When selecting the pressure value, it is important to take into account the type of matrix. Some matrices may protect microorganisms against high pressure, for example, if the food product is surrounded by a lipid layer and dissolved sugars or glycerol are present. The matrix also affects the way pressure is transferred within the product. It has been observed that, in products with reduced water activity, the elimination of microorganisms is less effective, in particular due to the lack of a continuous phase transmitting pressure. It is also worth emphasizing that, during pressure, the texture may change, which may result in, for example, water leakage or a change in pH [54,59]. The impact of HHP on food microorganisms may be increased if natural bactericides are present in the matrix or have been intentionally added. Higher inactivation can be observed in products with a lowered pH; hence, identical pressure parameters in different matrices may induce different elimination for the same microorganisms [3]. The above-mentioned observations confirm the problem of HHP food preservation using sufficiently high pressure parameters. It has been observed that, depending on the pressure used for germination, the sensitivity to other environmental stresses changes. In the research conducted by Wuytack et al. [60], it was observed that those spores germinating at a pressure of 100 MPa for 30 min were more sensitive to UV irradiation and the effect of hydrogen peroxide than the spores germinating at a pressure of 500 MPa. The method of applying pressure has a significant impact on achieving the level of microbial inactivation. Moreover, the use of cyclic pressure is more effective compared to the continuous method [43,61].

Temperature control during HHP processing is a key factor during food preservation. Typically, moderate or low temperatures are used when preserving food using the HHP technique. The combination of low temperature and high pressure can increase the efficiency of microorganism inactivation due to better pressure penetration inside cellular structures, which increases the damage and inactivation of the bacterial cell. Additionally, moderate temperatures can help to preserve the sensory quality, which often occurs as a result of the protein denaturation caused by high temperatures. However, it is difficult to inactivate spores using commercially available pressures (600 MPa) and low temperatures (e.g., 10 °C). In this case, better results can be obtained at a high temperature [5,62]. Spores are more resistant to environmental stress factors, while a higher temperature during HHP may increase their sensitivity to pressure, which leads to their effective elimination. In the research conducted by Al-Ghamdi et al. [63], it was observed that the use of a temperature of 122 °C during the pressure-assisted thermal sterilization (PATS) process resulted in the complete inactivation of the *Bacillus amyloliquefaciens* spores in purple potatoes puree samples (9 log reduction). It has also been observed that increasing temperatures results in better bacteria reduction results at lower pressure values. In another study conducted by Lenz et al. [56], at a pressure of 750 MPa and temperature of 75 °C, a reduction in the *Clostridium botulinum* spores by approximately 4.5 log was demonstrated, which corresponded to the reduction at the temperature of 90 °C and a pressure of 600 MPa (4.6 log reduction).

## 3. Bacteriophages—A Short History of Good Viruses

The discovery of penicillin in 1928 resulted in significant improvements in the quality and length of human life. This “miracle drug” with amazing properties for treating bacterial infections quickly spread around the world and was used on an increasing scale every year. Unfortunately, the overuse of both penicillin and other chemotherapy drugs in the subsequent years led to the development of bacterial resistance to antibiotics, which significantly reduced the patient’s chance of a quick recovery [64]. Currently, antibiotics are used not only in medicine but also in breeding animals for slaughter [65]. Antibiotic residues are detected in wide ranges of food products of animal and plant origin, feed, groundwater, and soil [66]. The long-term exposure of the human to antibiotics may lead to the weakening of the immune system, destruction of intestinal microbiota (dysbiosis), kidney problems, and even a carcinogenic factor. Currently, strategies to overcome antibiotic resistance are being implemented and are based on the alternative use of bacteriophages or their lytic enzymes in targeted therapy (so-called experimental phage therapy) [67].

Bacteriophages (called phages), known as bacterial viruses, are the most numerous particles on Earth and occur wherever there is a potential host—bacteria. The phage particle is called a virion [68]. Currently, bacteriophages are defined as bacterial viruses that are built with a protein or protein–lipid capsid that protects the genetic material. They do not have a cellular structure and do not reproduce outside the host [69]. Bacteriophages occur in various environments where bacterial cells are present, such as sewage and soil (which are the best sources of phage isolation), water, and food products. Their presence has also been observed in human vaccines, commercially available sera, the oral cavity (saliva and dental plaque), skin and hair, and the gastrointestinal tracts of humans and animals. The ubiquitous presence of bacteriophages is a natural mechanism enabling the maintenance of bacterial balance in the environment [70]. The first reports about bacteriophages were presented in 1896 by British chemist Ernest Hanbury Hankin, who noticed that cholera cases were sporadic among people bathing in the Ganges and Jumma Rivers in India. Two years later, in 1898, Russian bacteriologist Gamaleya and several other researchers observed a similar phenomenon in the case of the Gram-positive bacilli *Bacillus subtilis*. However, it was not until almost 20 years later that British bacteriologist Frederick Twort put forward the hypothesis that the antibacterial agent could be a virus. Two years later, in 1917, microbiologist Felix d’Herelle became interested again in bacteriophages and was the first to isolate phages, and then he used them to treat bacillary dysentery in children [71].

Depending on the type of genetic material, bacteriophages can be divided into containing double-stranded DNA (the largest group of bacteriophages), single-stranded DNA, or RNA. The bacteriophage genomes vary in size, from 2.5 kb to approximately 150 kb. The number of genes is approximately proportional to the size of the genome.

Due to the arrangement of the protein subunits, we distinguish helical viruses, in which the subunits are shifted relative to each other by a specific angle and have an elongated, rod-shaped structure. An example of such viruses is the M13 phage [72]. There are also viruses with an icosahedral structure, which are characterized by the presence of at least 60 identical elements in the form of a cube or icosahedron, with a double, triple, or quintuple symmetry axis. Viruses with this structure include coliphage ΦX174 [73]. The most complicated group is complex viruses, which combine the structures of icosahedral and helical viruses. They consist of a head and a tail [74]. Due to the length and nature of the tail, phages are divided into short ones (e.g., T7 phage), long contractile ones (e.g., Mu phage), and long non-contractile ones (e.g., λ phage) [75].

Most bacteriophages (over 90% of those known so far and characterized) show complex structures. The tail shows a more complicated morphology. It consists of a collar, neck fiber, tube, tail fibers, tail projections, and baseplate [74]. The neck fiber has a detailed regulatory function: it prevents phage adsorption on the surface of bacteria in unfavorable conditions (e.g., a low pH) by binding with the tail fibers and keeping the phage pulled upwards [76]. The end of the tail is the baseplate, which plays an important role during phage infection. The baseplate has built-in enzymatic proteins whose task is to create pores in bacterial cell covers [77]. The long tail fibers extending from the baseplate are used to recognize the localized receptors on target cells [78,79].

The initial stage of the phage replication cycle is infection. The first step in phage infection is reversible adsorption [80]. An important role in this process is played by the baseplate and tail fibers, which recognize specific receptors on the bacterial cell surface. Virtually all surface structures, such as lipopolysaccharides, teichoic acids, proteins, and even bacterial flagella, can serve as receptors [75,80]. Due to the differences in the structure and chemical composition of the cell wall of Gram-positive bacteria, the receptors may be main and side lipopolysaccharide chains and membrane proteins (e.g., transport channels), and in Gram-negative bacteria PG chains and teichoic acids (e.g., PGP (poly–[phosphate glycerol]) and GTA (poly–[glucose–N–acetylgalactosamine phosphate])) [81,82,83]. The secondary receptors in Gram-negative bacteria may be proteins anchored in PG or having a domain penetrating its layer (e.g., transmembrane protein OmpA) [84]. For this reason, bacteriophages are usually capable of attacking only one of the subtypes of bacteria. As mentioned above, some bacteriophages can recognize bacterial flagella, thanks to which they can gradually move toward the base of the flagellum to attach to the host cell surface [85,86]. There are also known phages (e.g., fAcM4 and fAcS2 targeting Gram-negative aerobic heterotroph *Asticcacaulis biprosthecum*) that attach to the flagellum using a special protein at the junction of the head and tail (thanks to which the tail is free and ready to attach to the bacterial surface), and, interestingly, they can move along the flagellum towards the cell surface. Once a bacteriophage binds to a specific receptor, irreversible binding occurs and the phage cannot leave the host surface even if the cell is damaged or dead [87].

The process of injecting a phage’s genetic material into the bacterial host cell is often supported by specialized phage enzymes, the so-called polysaccharide depolymerases [75,81]. These enzymes are located on the phage’s baseplate or its tail. Thanks to them, phages can more easily penetrate the biofilm structure and overcome the lipopolysaccharide layer of the bacterial cell wall, enabling the injection of genetic material through the bacteriophage tail. After the injection of the virion’s genetic material, the remaining structural elements of the phage remain outside the bacterial host cell [88].

Depending on the course of the replication cycle, bacteriophages can be divided into two main groups. The most desirable phages in food biopreservation are virulent (lytic) phages [81,89]. In the lytic cycle, the metabolism of bacteria is changed to the production of virions and lytic proteins in the host’s cell. The main lytic enzymes include endolysins, which cut the PG of the bacterial cell wall, and holins, i.e., special proteins that create tunnels in the cytoplasmic membrane [79,90]. Thanks to holins, endolysins can reach PG, which, consequently, leads to the breakdown (lysis) of the infected cell [91]. The second group of bacterial viruses are temperate (lysogenic) bacteriophages. The lysogenic replication cycle involves the injection of the phage’s genetic material into the cytoplasm of the bacterial cell and its integration with the bacterial genome. During bacterial division, the phage genome is replicated together with the host genome and passed on to the subsequent generations of bacteria. The dormant genetic material of a phage is called a prophage [92]. Most benign bacteriophages (e.g., phage *λ*) integrate the genome into the bacterial chromosome [93], while some (e.g., phage P1) function in the form of a plasmid, i.e., a DNA fragment independent of the bacterial genetic material [94]. Lysogenicity may last for many generations, but an external stimulus, such as some antibiotics or UV radiation, may initiate (induce) the transition of the prophage to the lytic cycle. In this case, the lytic cycle proceeds in a similar way to that of obligate virulent bacteriophages [78]. Recent research has confirmed that bacteriophages can communicate using short peptides released into the environment, which help to “decide” which replication cycle to choose [95].

Once the phage is isolated, whole-genome sequencing (WGS) and genome analysis are necessary. The analysis of the phage genome will confirm the nature of the replication cycle and detect the possible markers of phage lysogenicity, including genes encoding toxins or antibiotic resistance [96]. One of the key aspects of phage selection is the fact that they infect only the target group of microorganisms, leaving the intestinal microbiota intact. Therefore, an important step in bacteriophage characterization is to determine the range of bacterial hosts [97]. It is worth emphasizing that, before use, phages must be well-cleaned of other substances. For example, bacterial endotoxins can lead to serious side effects, burdening the liver and other internal organs, among others. An endotoxin is a strong stimulus for leukocytes, endothelial cells, and epithelial cells [98].

Due to the protein envelope of the virion, bacteriophages are unstable outside certain environmental conditions. High temperature, an acidic or alkaline environment, salt concentration, or UV light may significantly reduce their activity. Therefore, it is important to determine the stability of the viruses before using them [99,100]. For example, a very effective and virulent bacteriophage may be stable within a narrow range, which prevents its use in the food industry [76].

### 3.1. The Influence of HHP on the Phage Stability

The use of bacteriophages in the food industry is a relatively new concept of food preservation. There are several commercialized preparations based on lytic phages available on the market, targeting specific types or strains of food pathogens, including *Salmonella* spp., *L. monocytogenes*, enterohemorrhagic *E. coli* O157:H7, and *Shigella* spp. [101]. Over recent years, the number of regulatory approvals issued for bacteriophage preparations and their use to improve food safety have continued to increase. In 2016, the EFSA issued a report assessing the safety of the use and effectiveness of Listex^TM^ P100 from Micreos Food Safety in combating *L. monocytogenes*, and its effectiveness was confirmed in scientific research [102]. Many other phage preparations have a GRAS (Generally Recognized As Safe) status, an FDA (Food and Drug Administration) recommendation, and a kosher certificate [103]. In the European Union, these preparations are not approved for use in the food industry. Commercial phage agents are, in turn, used, among others, in Australia, Brazil, Israel, Canada, New Zealand, Switzerland, and the USA [104].

During evolution, bacteriophages have developed many specific molecular mechanisms. One of them is the possibility of trapping the phage’s genetic material in bacterial endospores and making it dormant until germination. After this stage, the lytic cycle continues and the bacteria are destroyed. The described mechanism confirms the possible effectiveness of using phages against spore-forming bacteria but also indicates the frequent occurrence of lysogenic phages targeting spore-producing bacteria [105]. Recent studies on the characterization and analysis of the genome of Alicyclobacillus phage KKP 3916 targeting *Alicyclobacillus acidoterrestris* strain KKP 3133 indicate that the phage is stable over wide ranges of pH values, temperatures, and UV irradiation and was effective in inhibiting the growth of bacteria. However, the genome analysis revealed that the phage is not strictly lytic, and, due to the potential for the transfer of resistant genes, it is not desirable in food preservation [99].

An effective method of eliminating spores may be the use of a selected cocktail of strictly lytic phages isolated from the environment, combined with the HHP technique. This approach will enable the reduction in the vegetative forms of bacteria and the germination of spores under the influence of pressure, which will then be eliminated by phage monoculture or a cocktail of various phages. To avoid re-multiplying bacteria from induced spores, bacteriophages can be added to the final product before HHP is used. In this case, the phages must be stable within the applied pressure range. Bacterial viruses can also be applied after HHP as additional protection against the target bacteria [106,107]. Combining these two methods can significantly improve the microbiological safety of food while minimizing the impact on food quality and nutritional value. In addition, the use of the pressure values and process digestion time required for spore germination contributes to lower production losses, indirectly saving energy. The effect of pressure on bacteriophages has not been fully explored, but the main mechanisms of the inactivation of bacterial viruses can be distinguished, which are presented in Figure 2.

Numerous studies confirm that the sensitivity of bacteriophages to HHP is related to their morphology (i.e., size, volume, or shape). Often, smaller viruses with an icosahedral virion show negligible susceptibility to HHP [106]. In the case of larger icosahedral viruses, the basis for inactivation is disruption of the structure of the virion and/or capsid. Often, the disrupted structure of the virion causes the release of the phage’s genetic material into the environment. Some bacteriophages are not destroyed by high pressure but lose their ability to infect. This is linked to changes in the capsid or tail proteins, which prevent adsorption to the bacterial host [107]. Additionally, the inactivation of bacteriophages under the influence of hydrostatic pressure largely depends on parameters such as pressure power, time, and temperature [108,109]. Inactivation also depends on the type of product, i.e., its pH, salt content, water activity, and storage temperature [99,100]. Table 3 presents a collection of studies on the influence of pressure parameters, temperature, and time exposure on HHP, and the type of matrix on the stability of bacteriophages. 

A review of the studies on the stability of bacteriophages under various HHP parameters confirms the possibility of successfully combining the two methods. It has been observed that bacteriophages with small genomes are more resistant to hydrostatic pressure [113]. A pressure of 400 MPa for bacteriophages with large genomes can cause complete inactivation. Due to the structure of the phage’s genetic material, DNA is more resistant to pressure compared to RNA, making DNA phages generally more stable. It is worth noting that the stability of bacteriophage viruses largely depends on the process parameters (pressure, time, temperature, and matrix) and the type of bacteriophage used.

### 3.2. Synergistic Effect of Phages and HHP in Bacterial Inactivation

The proper use of hurdle interventions makes it possible to effectively eliminate or achieve the reduction to low levels of pathogens in food matrices. Depending on the food composition, the combination of HHP and bacteriophages may not provide the expected food preservation results. In such a case, it is necessary to use other bacteria-eliminating factors, i.e., bacteriocins (e.g., nisin and lysozyme), essential oils, or organic acids (e.g., lactic acid). This combination can enable minimizing the operational parameters, thereby reducing the cost of production and maintaining a more natural product appearance. Table 4 summarizes the studies on the combination of hurdles (i.e., HHP and bacteriophages) and the synergistic effect on bacteria inactivation in various matrices.

In the study conducted by Komora et al. [115], the combination of HHP (300 MPa/5 min/10 °C) and Listex^TM^ P100 biopreparation in fermented sausage did not completely eliminate *L. monocytogenes*. In turn, after adding pediocin PA–1 produced by *Pediococcus acidilactici* strain HA 6111–2, in combination with HHP and bacteriophages, a reduction in this pathogen by 5 log CFU g^−1^ was achieved, which resulted in the complete elimination of the *L. monocytogenes* immediately after the treatment without re-growth during storage. The combination of two hurdles (i.e., HHP and pediocin PA–1) resulted in complete bacteria elimination in fermented meat sausage achieved after 72 h of storage. After the addition of pediocin PA–1 alone to the food matrix, no bacteria were detected only after the 21st day of storage. Similar results were obtained after the mere application of the Listex^TM^ P100 biopreparation. Moreover, a decrease in the phage titer was observed during storage, which was probably related to the acidification of the food product [115]. The effectiveness of the combined action depends on the temperature because high temperatures may cause denaturation of viral proteins, while low temperatures affect the enzymatic activity of bacteria, which consequently leads to reduced virion replication and their viability. Pereira et al.’s [108] study showed that, during cold storage, the number of bacteriophages in the presence of the host was similar compared to the control.

In some cases, it was observed that the use of bacteriophages or high-pressure treatment was ineffective in controlling the bacteria in food products. Combining both methods showed promising results. The structure of the product plays a significant role in the success of phage biocontrol [108,109,111]. In liquid products, such as juices, fruit drinks, smoothies, or purees, bacteriophages have a greater chance of encountering a bacterial host cell and infecting it compared to solid products. It was observed that products such as fish and mussels enable the use of lower pressure values while maintaining the microbiological safety of the product. Due to the small number of studies conducted on the use of this combined technology, it is impossible to determine in which food matrix the best results in combating the pathogen occur. Additionally, the lack of research on the impact of the synergistic effect of HHP and bacteriophages on spore-forming bacteria makes it impossible to accurately predict the effectiveness of this method.

## 4. Conclusions and Future Perspectives

High-hydrostatic-pressure technology is an innovative, non-thermal method of food preservation that is used commercially, especially in the juice industry. Bacterial cells exposed to HHP undergo both lethal and sublethal damage. Sublethally damaged cells may become healthy cells through regeneration and again pose a potential microbiological threat in food products. Moreover, a review of the previous research confirms the difficulties in eliminating bacterial spores, including from the *Alicyclobacillus*, *Clostridium,* and *Bacillus* genera, after using the HHP technique. Therefore, it is necessary to search for new or combine several different methods of food preservation (hurdle technology) to ensure the microbiological safety of food and protect consumers’ health. One of the increasingly considered methods of eradicating bacteria from the food industry environment is the use of strictly specific virulent bacteriophages. Phage cocktails can be used in three sectors of the food industry: (a) primary production, including animal breeding and field crops; (b) bio-sanitization, mainly in the production plants, to prevent the formation of biofilms on the surface of equipment; and (c) biopreservation, which aims to extend the shelf life of food by limiting the growth of saprophytic and pathogenic bacteria. An effective method of eliminating spore-forming bacteria may be the use of a selected cocktail of strictly lytic phages isolated from the environment, combined with the HHP technique. This approach will enable the germination of spores with the simultaneous reduction in vegetative bacterial cells, which will be eliminated as a result of lysis using bacteriophages applied to the food matrix. However, the challenge is to select strictly lytic and at the same time HHP-resistant bacteriophages. Future research should be aimed at investigating the potential mechanisms of the baroprotective activity related to food ingredients and properties (chemical compounds and/or ionic strength) that may have the ability to protect phages against HHP treatment.

## Figures and Tables

**Figure 1 foods-13-02519-f001:**
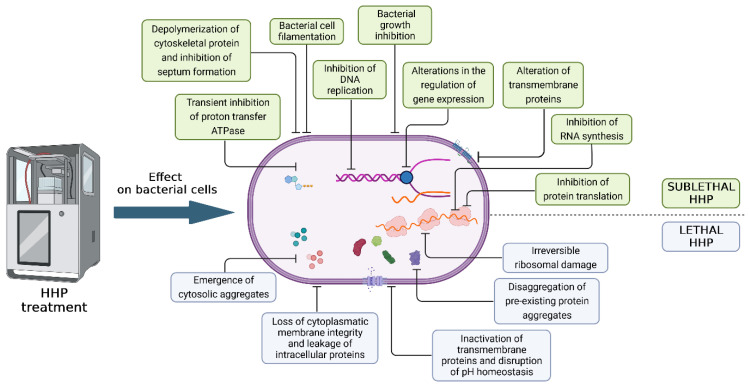
Damage to bacterial cells under the influence of high hydrostatic pressure. The figure was prepared in the BioRender program (license number: DQ267L317W).

**Figure 2 foods-13-02519-f002:**
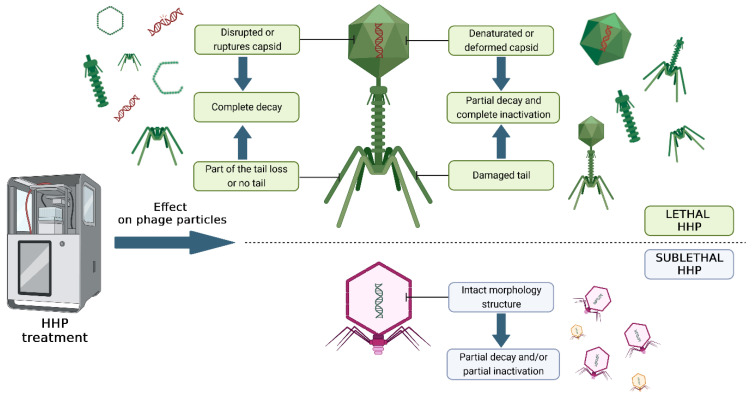
Damage to bacteriophage particles under the influence of high hydrostatic pressure. The figure was prepared in the BioRender program (license number: HY267L340M).

**Table 1 foods-13-02519-t001:** Reductions in the bacterial levels after HHP treatment at various pressure, time, and temperature parameters in food products.

Bacteria	Matrix	HHP Parameters			Inactivation (↓log CFU mL^−1^ or log CFU g^−1^)	Reference
		Pressure (MPa)	Time (min)	Temperature (°C)		
*Aeromonas hydrophila* strain AH 191	UHT whole milk	250	10	25	~2.5	[28]
		350	4		~7.0	
*Escherichia coli*	beetroot juice (pH 4.18)	400	10	20	6.2	[29]
*Escherichia coli*	skim milk	300	5	nd	2.1	[30]
		400			complete reduction	
*Listeria innocua*	beetroot juice (pH 4.18)	400	10	20	7.0 after 1 min	[29]
*Listeria monocytogenes*	cooked chicken	500	1	20	0.9	[31]
				40	3.8	
*Listeria monocytogenes*	skim milk	300	5	nd	1.5	[30]
		400			3.4	
		500			complete reduction	
*Salmonella* Enteritidis	liquid whole egg	200	10	nd	4.89	[32]
		300			~5.20	
		400			5.31	
*Salmonella* Typhimurium	skim milk	300	5	nd	2.8	[30]
		400			complete reduction	
*Staphylococcus aureus*	liquid whole egg	200	10	nd	1.84	[32]
		300			~2.00	
		400			2.63	
*Staphylococcus aureus*	skim milk	300	5	nd	0.50	[30]
		400			4.00	
		500			5.85	

Abbreviations: nd—no data.

**Table 2 foods-13-02519-t002:** Reductions in the levels of bacterial spores after HHP treatment at various pressure, time, and temperature parameters.

Bacteria	Matrix	HHP Parameters			Inactivation (↓log CFU mL^−1^ or log CFU g^−1^)	Reference
		Pressure (MPa)	Time (min)	Temperature (°C)		
*Alicyclobacillus acidoterrestris*	orange juice (pH 3.7, 11.45 °Brix)	600	5	60	3.0	[49]
			10		3.5	
*Alicyclobacillus acidoterrestris*	orange juice (pH 3.8, 9.20 °Brix)	600	10	45	~1.0	[50]
*Alicyclobacillus acidoterrestris*	apple juice (pH 3.4, 11.20 °Brix)	200	20	50	1.95	[43]
				70	3.99	
		500		70	6.13	
*Alicyclobacillus acidoterrestris*	tomato pulp (pH 4.2)	200	10	40	1.0–1.5	[51]
				60		
		400		40		
				60		
		600		40	to 3.5	
				60		
*Bacillus cereus*	CPB (a_w_ = 0.92)	600	5	70	~6.0	[52]
	CPB (a_w_ = 0.85)				3.0	
	CPB (a_w_ = 0.80)				1.5	
*Bacillus cereus*	MES buffer	600		100		
*Bacillus coagulans*	tomato pulp	300	15	50	2.0	[53]
				60	2.4	
		600		50	3.1	
				60	5.7	
*Bacillus coagulans*	tomato sauce (pH 4.2)	600	10	60	2.0	[51]
*Bacillus subtilis*	honey (a_w_ = 0.85)	600	15	85	0	[54]
*Bacillus subtilis*	distilled water	350	10	40	1.0	[55]
*Clostridium botulinum*	IPB (pH 7.0)	450	10	45	1.0	[56]
		600		75	2.1	
		750			5.6	
		900			complete reduction	
				30	3.8	
*Clostridium perfringens*	beef slurry (pH 6.5)	600	20	75	2.2	[57]
*Paenibacillus* sp.	UHT milk	500	10	20	0.5	[58]
				50	2.0	
		600		20	1.1	
				50	2.7	
*Terribacillus aidingensis*		500	10	20	2.1	
				50	2.2	
		600		20	2.3	
				50	2.2	

Abbreviations: CPB—citrate-phosphate buffer; MES buffer—2-morpholinoethanesulphonic acid buffer; IPB—imidazole-phosphate buffer.

**Table 3 foods-13-02519-t003:** The influence of HHP parameters (pressure, temperature, and time) on the bacteriophage activity in various matrices.

Phage	Type of Genetic Material, Genome Size (bp)	Bacterial Host	Matrix	HHP Parameters			Phage Inactivation (↓log PFU mL^−1^ or log PFU g^−1^)	Reference
				Pressure (MPa)	Temperature (°C)	Time (min)		
phiIPLA35	dsDNA 45,344	*Staphylococcus aureus*	pasteurized milk	0–400	10	5	no reduction	[109]
phiIPLA88	dsDNA 42,526			500			~1.0	
				600			~5.0	
				700			complete reduction	
phT4A	dsDNA 171,598	*Escherichia coli*	TSB	75	~21	5	0.29	[108]
						20	0.46	
						30	0.61	
				200		5	2.44	
						20	2.56	
						30	2.71	
				300		5	3.13	
						20	3.45	
						30	4.26	
				400		5	4.94	
						20	6.26	
						30	complete reduction	
Bacteriophages cocktail Salmonellex^TM^	dsDNA	*Salmonella* Typhimurium strain DT104	egg white	200	10	5	~0.5	[110]
				300			<1.0	
				400			1.7	
				500			3.4	
				600			5.5	
Bacteriophages cocktail Salmonellex^TM^	dsDNA	*Salmonella* Typhimurium strain DT104	egg yolk	200	10	5	~0.5	[110]
				300			<1.0	
				400			~1.0	
				500			3.4	
				600			5.9	
			liquid whole egg	200			~0.5	
				300			<1.0	
				400			~1.0	
				500			2.2	
				600			5.8	
			PBS	200			<1.0	
				300			1.2	
				400			3.9	
				500			7.0	
				600			6.8	
			phage stock solution	200			<1.0	
				300			<1.0	
				400			1.0	
				500			1.5	
				600			9.8	
phage P100	dsDNA 131,385	*Listeria monocytogenes*	fermented sausage “Alheira” (pH 6.07)	200	10	5	<0.5	[111]
				300			0.9	
				400			complete reduction	
			cheese (pH 5.66)	200			<0.5	
				300			0.9	
				400			complete reduction	
			PBS (pH 7.42)	200			<0.5	
				300			2.8	
				400			complete reduction	
			UHT milk (pH 6.73)	200			<0.5	
				300			0.8	
				400			complete reduction	
			apple juice (pH 3.41)	200			3.0	
				300			7.0	
				400			complete reduction	
			orange/carrot nectar (pH 3.54)	200			3.0	
				300			7.0	
				400			complete reduction	
φLd66-36	no data	*Lactococcus lactis* subsp. *lactis*	M17 broth	300	~25	10	~2.5	[112]
						20	~2.6	
						30	~3.9	
						40	~3.8	
						50	~4.7	
						60	~6.8	
φLd66-36	no data	*Lactococcus lactis* subsp. *lactis*	M17 broth	350	~25	5	~2.2	[112]
						10	~4.3	
						15	~5.0	
						20	~6.3	
φX174	ssDNA 5386	*Escherichia coli*	PBS	450	21	5	~0.7	[113]
						10	~0.5	
						15	~0.6	
						20	~0.6	
						60	0.8	
				600		5	~0.6	
						10	~0.5	
						20	~0.6	
						60	~0.7	
				350	21	5	~0.8	
				400			~0.7	
				450			~0.7	
				500			~0.4	
				550			~0.4	
				600			~0.6	
				600	4	5	~0.6	
					10		~0.6	
					21		~0.6	
					30		~0.5	
					40		~0.6	
MS2	ssRNA 3569	*Escherichia coli*	PBS	450	21	5	0.2	[113]
						10	~0.5	
						15	0.8	
						20	~1.1	
						60	~1.1	
				600		5	~3.4	
						10	~3.5	
						20	~3.7	
						60	4.0	
				350	21	5	0.5	
				400			~0.3	
				450			~0.2	
				500			~0.7	
				550			1.5	
				600			~3.4	
				600	4	5	4.0	
					10		4.4	
					21		~3.4	
					30		~3.4	
					40		~3.4	
ϕAbc2	dsDNA 34,882	*Streptococcus thermophilus*	M17 broth	400	~33	5	~0.6	[114]
						10	~1.4	
						20	~2.4	
						30	~3.0	
				500	~35	5	~5.1	
						10	~5.4	
						20	~7.3	
				600	~38	5	complete reduction	
ALQ13.2	dsDNA 35,525			400	~33	5	~0.8	
						10	~1.2	
						20	~1.7	
						30	~2.0	
				500	~35	5	~1.2	
						10	~2.1	
						20	~2.0	
						30	~2.8	
				600	~38	5	3.7	
						10	~6.0	
						20	~7.8	
DT1	dsDNA 34,820	*Streptococcus thermophilus*	M17 broth	400	~33	5	~0.6	[114]
						10	~1.4	
						20	~2.6	
						30	~3.6	
				500	~35	5	~4.9	
						10	~6.4	
						20	~8.3	
				600	~38	5	complete reduction	

Abbreviations: TSB—tryptic soy broth; PBS—phosphate-buffered saline.

**Table 4 foods-13-02519-t004:** Synergistic effect of HHP and bacteriophage treatment on bacteria inactivation in various matrices.

Bacterial Host (Contamination Level)	Matrix	Phage	HHP Parameters			Storage		Inactivation (↓log CFU mL^−1^ or log CFU g^−1^)	Reference
			Pressure (MPa)	Temperature (°C)	Time (min)	Temperature (°C)	Time (Day or Hours)		
*Listeria monocytogenes* strain ScottA (5.0 log CFU g^−1^)	fermented meat sausage “Alheira” (pH 6.07)	Listex™ P100	300	10	5	4	0 h	3.2	[111]
							14 d	3.2	
							60 d	3.2	
		–					7 d	0.3	
							14 d	0.3	
							60 d	0.3	
*Listeria monocytogenes* strain 1942 (5.0 log CFU g^−1^)		Listex™ P100					0 h	3.2	
							14 d	3.2	
							60 d	3.2	
		–					7 d	0.3	
							14 d	0.3	
							60 d	0.3	
*Shigella flexneri* (3.9 log CFU g^−1^)	ground beef	AG10	250	~25	5	~21	0 h	complete reduction	[106]
		–						2.5	
*Staphylococcus aureus* (4.0 log CFU mL^−1^)	pasteurized whole milk	phiIPLA35 phiIPLA88	400	10	5	25	4 h	~2.0	[109]
							1 d	complete reduction	
							2 d	complete reduction	
		–					2 h	~2.0	
							1 d	~1.5	
							2 d	~1.5	
*Vibrio cholerae* (3.8 log CFU g^−1^)	mussels	JA-1	250	~25	5	~21	0 h	2.6	[106]
			300		13			complete reduction	
		–	250		5			2.2	
			300		13			complete reduction	
	salmon fillet	JA-1	250		5			2.3	
			300		13			complete reduction	
		–	250		5			2.1	
			300		13			complete reduction	

## Data Availability

No new data were created or analyzed in this study. Data sharing is not applicable to this article.

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
