# Peer review of "The Novel Concept of Synergically Combining: High Hydrostatic Pressure and Lytic Bacteriophages to Eliminate Vegetative and Spore-Forming Bacteria in Food Products"

_foods, 2024, doi:10.3390/foods13162519_

Round 1

Reviewer 1 Report

Comments and Suggestions for Authors

The present review article involves the combined use of high hydrostatic pressure and lytic bacteriophages to reduce the formation of sporing bacteria in foods. The topic addressed is novel as limited studies are available reporting the combined use of these non-thermal and biochemical methods.

The review is well organized and provides the literature with a collective study and future perspectives. The authors used 115 references indicating the in-depth organization of this study.

The topic falls within the aims and scope of Foods journal and is important for Food Scientists.

In my opinion, the authors must report more details on the origin of these lytic bacteriophages, their properties in different foods, if adopted, and how their exploitation can be achieved. For example, methods of isolating these bacteriophages. These can also be included in Tables. I did not see it well in the review. In addition, in Figure 2 reporting the damages to bacteriophages particle after the application of high hydrostatic pressure, some schemes refer to insects. Can beneficial bacteriophages be isolated from insects? Do the authors know any relevant literature? Finally, the authors statement about the most relevant foods that this combination can be applicated must be given. Is only juices? meat? poultry? etc.

Other comments: Line 30 and elsewhere: Change ''place'' to ''give''.

Based on the overall input of this review, I suggest a minor revision.

Comments on the Quality of English Language

The English language is at good level. Minor changes are required.

Reviewer 2 Report

Comments and Suggestions for Authors

The problem of this review is its ambiguity. It could well pass in the current form as a review for HPP and bacteriophages in general, without the accent to spore-forming bacteria. However, for a review focused on spore-forming bacteria, the sources do not cover adequately the topic. E.g., the literature review on non-spore-forming bacteria and HPP (Table 2) do not include many important sources (e.g. https://doi.org/10.3390/foods13121832 ).

 Similarly, the review is missing studies on bacteriophage treatment of foodstuffs involving spore-forming bacteria (e.g.https://doi.org/10.1016/j.foodcont.2023.110157, and others.

So I suggest to either officially unfocus on spore-forming bacteria, or to focus more on them.

Aside from that, the review is well-written.

I detected only several small issues:

L169: which leads to its condensation

L250: I suggest to keep with "cortex" instead of "bark"

L410-411: please remove: (Ramisetty and Sudhakari 2019) = reference [92]

L445: Micreos

Reviewer 3 Report

Comments and Suggestions for Authors

Foods-3135986

The present review explains very well how high hydrostatic pressure (HHP) affects bacteria. Due to resistance of spore-forming bacteria against HHP, the authors present the benefits of applying hurdle technologies to control foodborne pathogens and spoilage microorganisms in food products. One alternative to improve HHPP efficiency is its combination with bacteriophages. Before introducing the scientific papers that present the efficacy of combine HHP with bacteriophage, the authors explain in detail the types and classification of bacteriophage. However, only scientific research about their use against vegetative cells were presented, and authors acknowledged this. For this reason, the title could be expanded to include that it is also about vegetative bacteria. I would like to note that authors provided an organized document that included some recent research paper.

Below are some recommendations to improve the quality of the manuscript.

Line 66: In Table 2 and Table 4 you provide a lot of information about vegetative cells, then the aim of your manuscript could be modified to “This article aims to elucidate the novel concept of synergically combining HHP and lytic bacteriophages to eliminate vegetative and spore-forming bacteria from food products effectively.” In addition, also in the title.

Line 197 (Table 1). Because table 1 is divided, the columns headings are needed in the second part to understand the information. Also, if the information from the same paper (in this case, ref. 24) appears in the second part of the table, include all the HHP parameters to properly follow the information. This issue also occurs with the other tables. It would be interesting to complete the characteristics of the food matrix in all studies, where possible, because, as you mentioned in the text, endogenous factors have a significant effect on microorganism and the effectiveness of HHP.

Table 1 and Line 235 to 238. Why did you not include the information from reference 36 in the table? I propose to include it in the table because authors studied the survival of L. innocua in a food product. Remove the information about the vitro results (PBS) of reference 29 from the table. Then, the title of the table can be modified to: “Reduction of the bacterial level after HHP treatment, at various pressure, time, and temperature parameters in food products.”

Line 204. Include reference about: “microorganism can also be achieved at negative temperatures.”

Line 261. Correct Ca-DPA to CaDPA (as in line 249 and 255)

Line 268 (Table 2). Because table 2 is divided, the columns headings are needed in the second part to understand the information. It would be interesting to complete the characteristics of the food matrix in all studies, where possible. For example, in reference 54, which worked with honey - a very different matrix compared to the others- knowing the water activity or solid soluble content could be valuable.

Line 270. Added the meaning of MES

Line 278. You can include the results observed in reference 54, where no reductions were observed when the soluble solids were high (84,6 °bx), but significant differences were noted when the honey was diluted.

Lines 293, 311, 316.  To include the reference to the paper you mention once is enough.

Line 349. Added reference 71 at the end of the paragraph.

Line 382. A parenthesis was missed after phosphate]))

Line 410. Correct the way of citation because it is reference 92.

Line 494. After this page (13 of 28), the page number should be corrected in the rest of the document.

Line 494 (Table 3). Since the table is divided in the document, each page should have the columns headings to understand the information.

-          Table 3. On page 15, for reference 110, the type of genetic material has been omitted; if no data is available, write ‘no data’ as in reference 112 (Table 3).

-          Place all the information for reference 111 on the same page or rewrite the important information in the first columns to make the results understandable.

-          The same issue occurs on pages 16, 17, 18, and 19, where some important information does not appear with the inactivation results.

Line 499. You can include the reference 113 to justify your phrase.

Line 550. You can include reference 108, 109, and 111 to justify your phrase, according to the results presented in Table 3.

Line 515 (Table 4). Divide the column for HHP parameters and storage, as the current format is confusing.   Be careful with the information from reference 109 because it appears divided in both pages; include all of it on one page. Also, add column heading in part 2.

Line 522. The authors evaluated this in a fermented sausage, which is a solid product, so change mL to g.

Lines 526, 535.  To include the reference to the paper you mention once is enough.
